# Dynamic [18]F-Pretomanid PET imaging in animal models of TB meningitis and human studies

Filipa Mota [1,2,3,7], Camilo A. Ruiz-Bedoya[1,2,3,7], Elizabeth W. Tucker [1,2,4,7], Daniel P. Holt[5,7], Patricia De Jesus [1,2,3], Martin A. Lodge[5], Clara Erice[1,2,4], Xueyi Chen [1,2,3], Melissa Bahr [1,2,3], Kelly Flavahan[1,2,3], John Kim [1,2,4], Mary Katherine Brosnan[5], Alvaro A. Ordonez [1,2,3], Charles A. Peloquin[6], Robert F. Dannals[5] & Sanjay K. Jain [1,2,3,5] ✉

Pretomanid is a nitroimidazole antimicrobial active against drug-resistant *Mycobacterium tuberculosis* and approved in combination with bedaquiline and linezolid (BPaL) to treat multidrug-resistant (MDR) pulmonary tuberculosis (TB). However, the penetration of these antibiotics into the central nervous system (CNS), and the efficacy of the BPaL regimen for TB meningitis, are not well established. Importantly, there is a lack of efficacious treatments for TB meningitis due to MDR strains, resulting in high mortality. We have developed new methods to synthesize [18]F-pretomanid (chemically identical to the antibiotic) and performed cross-species positron emission tomography (PET) imaging to noninvasively measure pretomanid concentration-time profiles. Dynamic PET in mouse and rabbit models of TB meningitis demonstrates excellent CNS penetration of pretomanid but cerebrospinal fluid (CSF) levels does not correlate with those in the brain parenchyma. The bactericidal activity of the BPaL regimen in the mouse model of TB meningitis is substantially inferior to the standard TB regimen, likely due to restricted penetration of bedaquiline and linezolid into the brain parenchyma. Finally, first-in-human dynamic [18]F-pretomanid PET in six healthy volunteers demonstrates excellent CNS penetration of pretomanid, with significantly higher levels in the brain parenchyma than in CSF. These data have important implications for developing new antibiotic treatments for TB meningitis.

Tuberculosis (TB) remains one of the leading killers from a single infectious agent[1] and TB meningitis is the most devastating extra-pulmonary form, especially in the young and immunocompromised[2–4]. Multidrug-resistant (MDR)-TB, caused by *Mycobacterium tuberculosis* resistant to first-line antibiotics (i.e., isoniazid and rifampin), is on the rise. TB meningitis due to MDR strains is associated with high mortality[5–7], and drug-resistance is an independent predictor of death[8]. In a recent retrospective cohort study among 237 patients with TB

meningitis, mortality was significantly higher among patients with drug-resistant (67%) than drug-susceptible disease (24%, *P* < 0.001)[9]. Moreover, mortality was significantly higher (adjusted hazards ratio of 7.2) in patients with drug-resistant TB meningitis after 90 days of initiation of treatment (*P* < 0.001). New drugs and more efficacious treatments against MDR-TB are therefore urgently needed to combat this public health threat. Pretomanid (formerly PA-824) is a small molecule belonging to the nitroimidazole class of antimicrobial

agents, approved by the U.S. Food and Drug Administration (FDA) in 2019 for the treatment of pulmonary MDR-TB, in combination with bedaquiline and linezolid (BPaL - bedaquiline, pretomanid, linezolid)[10]. Pretomanid is active against both replicating and non-replicating *M. tuberculosis*, which contributes to its excellent bactericidal activity[11–14].

With few exceptions, current antibiotic dosing recommendations are based on plasma concentrations, without information on drug concentrations at the site of infection. Since inappropriate antibiotic levels in target tissues can lead to selection of resistant organisms, toxicity or organ injury, and ultimately treatment failure, a growing number of studies and the U.S. FDA increasingly support measuring antibiotic concentrations in infected tissues[15]. Therefore, we have developed positron emission tomography (PET)-based clinically-translatable technologies for noninvasive, simultaneous, and unbiased, multi-compartment in situ measurements of antibiotic concentration-time curves in animals and humans[16–18]. In this study, we report the development of [18]F-pretomanid as a molecular imaging tool to noninvasively assess whole-body drug biodistribution (Fig. 1) utilizing detailed animal studies in mouse and rabbit models of TB meningitis[17,19,20]. In brief, infected animals undergo dynamic PET/computed tomography (CT) with [18]F-pretomanid to obtain time-activity curves (TACs) and areas under the curve (AUCs) by quantifying the PET signal in volumes of interest (VOI). Postmortem autoradiography and histology are also performed in all animal models. Given the unknown potential of pretomanid-containing regimens for TB meningitis, the bedaquiline (B), pretomanid (P) and linezolid (L) regimen is tested in a mouse model of TB meningitis[20]. Mass spectrometry and traditional microbiology techniques are employed to evaluate intraparenchymal drug levels and bactericidal efficacy (bacterial burden quantified with colony-forming units [CFU]) longitudinally. Radiosynthesis of [18]F-pretomanid under current Good Manufacturing Practices (cGMP) facilitates translation into humans, and first-in-human dynamic [18]F-pretomanid PET studies are performed in accordance with U.S. FDA guidelines.

## Results

### Radiosynthesis and characterization of [18]F-pretomanid

[18]F-Pretomanid was obtained through halogen exchange [18]F-fluorination (Fig. 2) of an aryl-bromodifluoromethoxyl precursor, which was obtained from a five-step synthetic route (Figs. S1–2). The radiosynthetic approach employed a silver-catalyzed nucleophilic displacement, following an adapted protocol by Khotavivattana et al.[21]. We obtained a radiochemical yield (RCY) of $5 \pm 2\%$ (non-decay corrected [n.d.c.]), from starting [18]F-fluoride to formulated [18]F-pretomanid, and found that reducing the amount of precursor or changing the solvent had a detrimental effect on the RCY. High-performance liquid chromatography (HPLC) analysis on the isolated product revealed a radiochemical purity >98%. While this method allowed the radiosynthesis of [18]F-pretomanid for animal studies, the use of dichloroethane precluded its clinical translation. Therefore, we tested alternative reaction solvents to translate the synthesis of [18]F-pretomanid under cGMP conditions which unfortunately significantly reduced RCY. However, automated cGMP synthesis of [18]F-pretomanid was successful under microwave irradiation at 100 watts for 10 min (reaching 120 °C) in dimethylformamide in the absence of silver salts, despite previous reports that catalyst-free thermal activation had been unsuccessful for the preparation of [18]F-labeled aryl-OCF$_3$ compounds. Under these conditions, [18]F-pretomanid was obtained in $5.7 \pm 0.3\%$ n.d.c. yield and a specific activity of $68 \pm 2$ GBq/μmol. HPLC analysis showed ≥95% radiochemical purity and a single peak corresponding to the [19]F-reference pretomanid in the UV chromatogram (Fig. S3).

In vitro, unbound [18]F-pretomanid remained stable (>90%) in mouse, rabbit, and human serum at 37 °C for three hours. Defluorination was not observed. Pretomanid is known to be highly protein bound (-86%) in human plasma[22]. When incubated with mouse, rabbit, and human serum at 37 °C, the protein binding level of [18]F-pretomanid was 75–77% in healthy human, 78–80% in healthy rabbit, 80–83% in *M. tuberculosis*-infected rabbit, 75–80% in healthy mouse, and 74–78% in *M. tuberculosis*-infected mouse sera (Table S1). Overall, [18]F-pretomanid

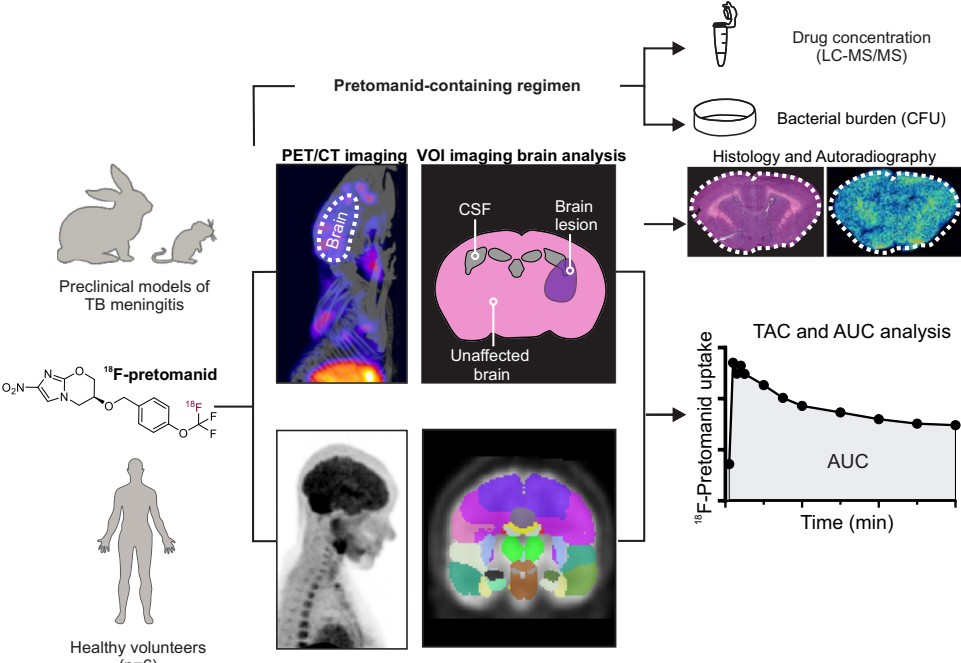

**Fig. 1 | Schematic representation of the experimental design.** Mouse and rabbit models of TB meningitis were used for radiotracer validation, pharmacokinetic characterization and pretomanid-containing regimen testing. [18]F-Pretomanid synthesis was optimized for clinical translation under cGMP and first-in-human PET imaging was performed. [18]F-Pretomanid was administered intravenously. Following dynamic PET/CT imaging, several volumes of interest (VOI) were drawn to obtain PET-derived time-activity curves (TACs). TACs were used to calculate the area under the curve (AUC) and AUC tissue/plasma ratios.

**Fig. 2 | Synthesis of <sup>18</sup>F-pretomanid.** ¹⁸F-pretomanid was obtained through ¹⁸F-fluorination of an aryl-bromodifluoromethoxyl precursor. Manual synthesis was performed for preclinical imaging studies and optimized to comply with cGMP to allow clinical translation.

had similar protein binding (74–83%) to unlabeled pretomanid, and no significant differences were found between species over time. ¹⁸F-Pretomanid experimental $LogD_{7.4}$, which represents its distribution coefficient at physiological pH, was $1.9 \pm 0.1$, which is only a 0.4 Log decrease when compared to unlabeled pretomanid[23]. Thus, ¹⁸F-labeled and unlabeled pretomanid are expected to have similar tissue partitioning. Whole-body biodistribution of ¹⁸F-pretomanid was measured in mice with experimentally-induced pulmonary TB utilizing PET/CT and gamma counting (Fig. S4). Upon intravenous injection, ¹⁸F-pretomanid rapidly distributed to all major organs, which was also confirmed by post-mortem biodistribution quantification by gamma-counting (Fig. S4a, b). The activity in the bone was low and did not substantially increase over time which indicates that defluorination did not occur in vivo (Fig. S4c). Similar to the parent drug, ¹⁸F-pretomanid underwent both renal and hepatobiliary excretion (Fig. S4d). Low uptake was observed in muscle and high uptake was found in brown adipose tissue (BAT), which cleared within hours (Fig. S4e). Spatial distribution with ex vivo autoradiography in the mouse model of pulmonary TB showed reduced uptake of ¹⁸F-pretomanid in lung lesions compared to unaffected lung (Fig. S4f). The upper-body biodistribution of ¹⁸F-pretomanid was also measured in rabbits showing similar findings as in mice (Fig. S5).

### ¹⁸F-Pretomanid has excellent central nervous system (CNS) penetration
Our initial studies characterizing the whole body biodistribution of ¹⁸F-pretomanid in mouse models of pulmonary TB revealed that the penetration into the brain parenchyma was high, with a median AUC ratio (brain/plasma) of 1.73 (IQR, 1.41–2.04), even in healthy brains. This prompted detailed investigation of ¹⁸F-pretomanid penetration into infected brains in two mammalian models of TB meningitis also showing excellent brain penetration of ¹⁸F-pretomanid [AUC ratios (brain/plasma) >1]. However, in vivo 3D PET/CT and 2D ex vivo autoradiography in the mouse model of TB meningitis showed reduced uptake of ¹⁸F-pretomanid with filling defects at the center of the brain lesion (visible in live animals with ¹⁸F-FDG PET/CT and ex vivo histopathology, respectively) (Fig. 3a, b). AUC ratio (brain/plasma) was 1.35 (median; IQR, 0.81–1.52) in brain lesions and 1.56 (median; IQR, 1.22–1.69) in unaffected brain regions (Fig. 3c, d). Similar findings were noted in a rabbit model of TB meningitis (Fig. 3e–h), with median AUC ratio (brain/plasma) of 1.87 (IQR, 1.66–4.63) into brain lesions and 2.75 (IQR, 1.64–5.73) into the unaffected brain.

### Assessment of the BPaL regimen in the mouse model of TB meningitis
We measured the efficacy of the pretomanid-containing BPaL regimen, currently the only U.S. FDA approved regimen for MDR-TB, in the mouse model of TB meningitis and compared it with the first-line, standard TB treatment (standard-dose rifampin, isoniazid, and pyrazinamide – $HR_{10}Z$) (Fig. 4a). Adjuvant dexamethasone was administered with both regimens per current TB meningitis treatment guidelines[2,24]. While both regimens decreased the bacterial burden compared to untreated mice, the bactericidal activity of the BPaL regimen was substantially inferior to the standard TB regimen ($P < 0.001$) (Fig. 4b–d and Table S2).

In previous studies[18,25], we have reported that the AUC ratios (brain/plasma) of ⁷⁶Br-bedaquiline and ¹⁸F-linezolid (both chemically identical to the parent drugs) into uninfected brain tissues were 0.15 and 0.34 respectively (Fig. S6). However, our current study demonstrated that ¹⁸F-pretomanid AUC ratios (brain/plasma) were high (>1.5) at the start of treatment (and remained high (>1) after two weeks of treatment (Fig. 4e). Brain parenchymal and CSF drug and metabolite levels were also measured by mass spectrometry, which demonstrated discordant penetration into the brain parenchyma and CSF compartments ($P = 0.002$, Fig. 4f–h and Table S3). While linezolid levels were higher in the CSF compared to the brain parenchyma, both pretomanid and bedaquiline levels were higher in the brain parenchyma compared to the CSF. Bedaquiline rapidly undergoes *N*-demethylation in vivo to form a metabolite (M2), which is also active against *M. tuberculosis*. We found that M2 levels (albeit still low) were higher in the brain parenchyma than the parent drug (Table S3).

It has been hypothesized that outcomes in TB meningitis are also associated with changes in intracerebral inflammation[2,5]. Therefore, we also assessed levels for select inflammatory cytokines in brain lysates, which did not show significant differences between the two regimens (Fig. S7).

### First-in-human ¹⁸F-pretomanid PET studies
Six healthy volunteers (three male and three female), with an age range of 20–53 years were recruited in an ongoing first-in-human ¹⁸F-pretomanid PET study at the Johns Hopkins Hospital (Table S4). While this study was not designed to assess safety the procedures were safe, no adverse or clinically detectable pharmacological effects were noted and no anatomic abnormalities were detected in any subject. Similar to the animal studies, after intravenous administration, ¹⁸F-pretomanid rapidly distributed to all major organs, and underwent both renal and hepatobiliary excretion. Importantly, ¹⁸F-pretomanid demonstrated excellent penetration into the brain parenchyma and the CSF, with an AUC (tissue/plasma) ratio >1 (Fig. 5 and Fig. S8). Additionally, and similar to the findings in the animal models, ¹⁸F-pretomanid exposures were compartmentalized with significantly lower penetration noted in the CSF (ventricles), compared to the brain parenchyma (Fig. 5; $P = 0.018$).

## Discussion
While optimal drug dosing can shorten current treatment regimens, suboptimal dosing is a major factor promoting antibiotic resistance, which the World Health Organization declared as one of the top ten threats to human health[26]. Notably, TB meningitis due to MDR strains lacks highly efficacious treatments and is associated with 40% to 100% mortality with current regimens[5–7,9]. Therefore, there is an urgent need

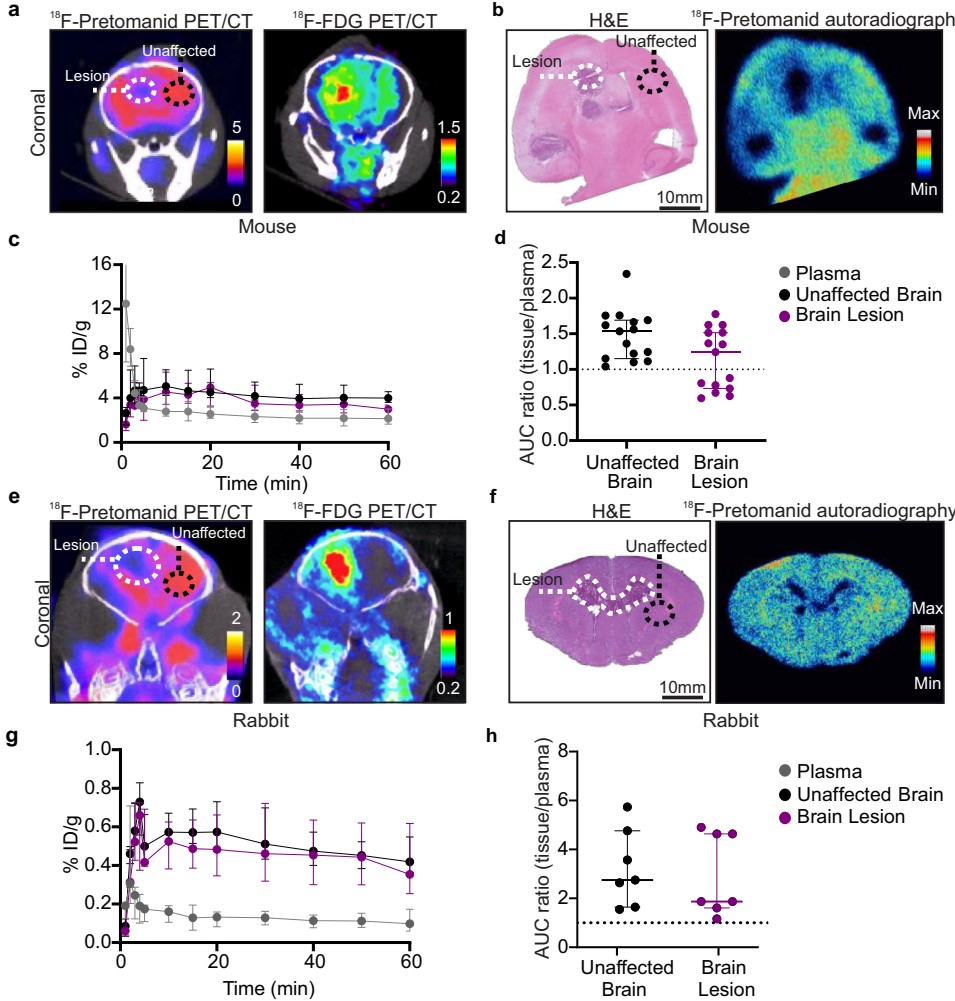

**Fig. 3 | Brain penetration of [18]F-pretomanid in mouse and rabbit models of TB meningitis. a–d** Mice, *n* = 8. **a** Coronal [18]F-pretomanid and [18]F-FDG PET/CT showing the brain lesion and unaffected brain. **b** Histopathology and [18]F-pretomanid autoradiography of the same mouse brain showing brain lesions (white dotted line) and unaffected brain (black dotted line). **c** Time-activity curves (TACs) from 0 to 60 min for plasma, brain lesions and unaffected brain. **d** Area under the curve (AUC) ratios (tissue/plasma) in unaffected brain [*n* = 15 volume of interests (VOIs)] and brain lesion (*n* = 15 VOIs) (*P* = 0.055). Each dot represents a VOI. **e–h** Rabbits, *n* = 4. **e** Coronal [18]F-pretomanid and [18]F-FDG PET/CT in the same rabbit. **f** Histopathology

and [18]F-pretomanid autoradiography of the same rabbit brain showing brain lesions (white dotted line) and unaffected brain (black dotted line). **g** TAC from 0 to 60 min for plasma, unaffected brain, and brain lesions. **h** AUC ratio (tissue/plasma) comparing unaffected brain (*n* = 7 VOIs) and brain lesions (*n* = 7 VOIs). Each dot represents a VOI. PET studies are based on microdoses (ng-µg) administered intravenously. Data are represented as median ± IQR. Statistical comparisons were made using a two-tailed Mann–Whitney-Wilcoxon test. Source data are provided as a Source Data file.

to optimize antibiotic treatments for TB meningitis, especially for those caused by MDR strains.

Pretomanid has been studied for pulmonary TB, and exhibits time-dependent bactericidal activity in the lungs[22]. However, there is limited data on its penetration into privileged sites such as the CNS[2]. Additionally, TB treatments can be particularly challenging because of the simultaneous co-existence of heterogeneous lesions with a complex microenvironment in the same host[16,17,20,27]. For instance, the blood-brain barrier (BBB) limits the access of several antimicrobials into the CNS[2,20]. Although mass spectrometry studies have demonstrated good CNS penetration of pretomanid in uninfected, healthy rats[28], these data provide measures only at a single time point and do not provide detailed spatiotemporal concentration-time measures in infected tissues. In addition to the unique microenvironments at the infection site, the physicochemical properties of specific drugs affect pharmacokinetic (PK) parameters such as drug penetration and clearance[29]. Our studies demonstrate that pretomanid has excellent penetration into the brain, including infected brain tissues [(brain/plasma) ≥1] in both mice and rabbits. While prior studies have shown

that rifampin penetration into the brain decreases dramatically as early as two weeks after initiation of TB treatments[17,20], we show that CNS penetration of pretomanid remains high [AUC (tissue/plasma) ratio >1] even after initiation of treatment with dexamethasone-containing regimens, which decreases BBB permeability.

Given the high brain penetration of [18]F-pretomanid, we measured the efficacy of the BPaL regimen, currently, the only U.S. FDA approved regimen for MDR-TB, in the mouse model of TB meningitis and compared it with the first-line, standard TB treatment (HR[10]Z). Adjuvant dexamethasone was administered with both regimens per current TB meningitis treatment guidelines. While the BPaL regimen decreased the bacterial burden in the brain (compared to untreated mice), surprisingly and contrary to what is noted for pulmonary TB[30], the bactericidal activity of the BPaL regimen was substantially inferior to the standard TB regimen, even six weeks after treatment initiation.

To understand the lower than expected efficacy of the BPaL regimen, we measured the penetration of each antibiotic into the CNS as single time-point post-mortem levels using mass spectrometry and measured pretomanid AUC in live animals using PET. As noted,

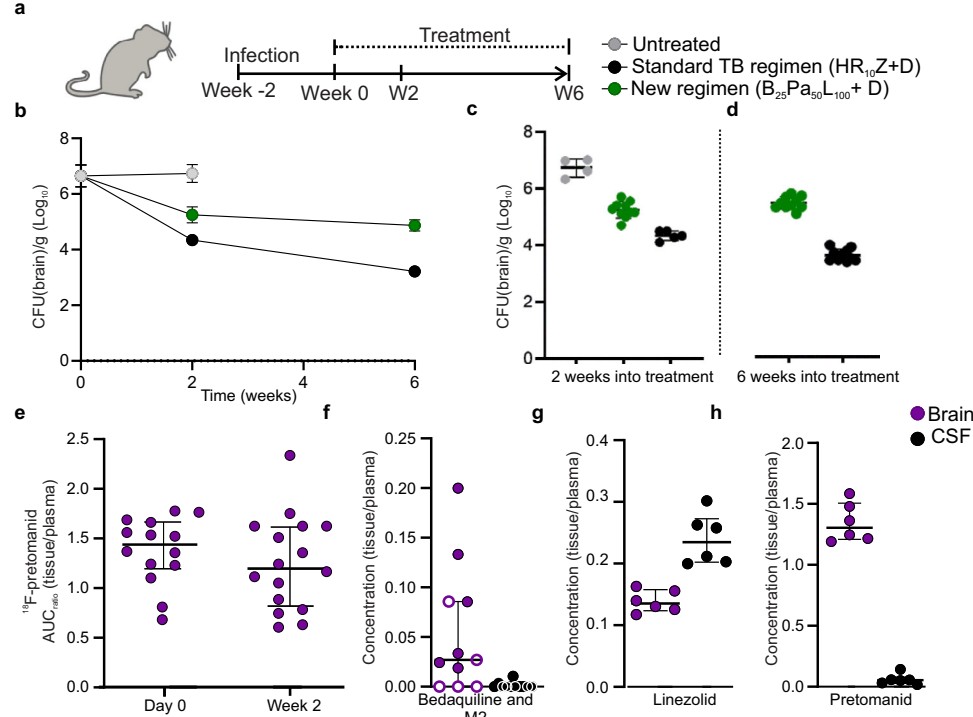

**Fig. 4 | Evaluation of BPaL regimen in the mouse model of TB meningitis. a** After two weeks of infection incubation (week 0), mice were randomly allocated into treatment groups; standard TB regimen (rifampin [R], isoniazid [H] and pyrazinamide [Z]), and a new regimen (BPaL; bedaquiline [B], pretomanid [P] and linezolid [L]). Rifampin (10 mg/kg/day), isoniazid (10 mg/kg/day), pyrazinamide (150 mg/kg/day), pretomanid (50 mg/kg/day divided twice daily), bedaquiline (25 mg/kg/day), and linezolid (100 mg/kg/day divided twice daily) were administered via oral gavage. Mouse dosing was utilized to match the standard human equipotent dosing: rifampin (10 mg/kg/day), isoniazid (10 mg/kg/day), pyrazinamide (25 mg/kg/day), pretomanid (200 mg/day), bedaquiline (standard oral dosing), and linezolid (1200 mg/day). All regimens received adjunctive dexamethasone via intraperitoneal injection. Four mice remained untreated for two weeks. **b** Bacterial burden over the treatment duration and (**c**) after two (*P* = 0.001)

and (**d**) six weeks (*P* < 0.001) of treatment (animal numbers at two weeks, *n* = 4/untreated, 5/standard TB regimen, and 10/BPaL; at six weeks *n* = 10/each group). **e** $^{18}$F-Pretomanid AUC ratios (brain/plasma) in mice with TB meningitis (*P* = 0.294). PET studies are based on microdoses (ng-µg) administered intravenously. Mass spectrometry-derived brain/plasma (purple dot) or CSF/plasma (black dot) concentration ratios for (**f**) bedaquiline [and M2 metabolite (open dot)] (*P* = 0.002), (**g**) linezolid (*P* = 0.002) and (**h**) pretomanid (*P* = 0.002) (*n* = 6/group) in mice with TB meningitis, following a single-dose and measured at T$_{max}$. CFU data is represented as mean ± SD on a logarithmic scale. PET and mass spectrometry data are represented as median ± IQR. Statistical comparisons were made using two-tailed Mann–Whitney-Wilcoxon test or a two-way ANOVA followed by Bonferroni's test. Source data are provided as a Source Data file.

pretomanid has excellent penetration into the CNS. However, in previous studies utilizing $^{18}$F-linezolid and $^{76}$Br-bedaquiline (both chemically identical to the parent drugs)[18,25] we have demonstrated that linezolid levels were lower than anticipated based on other published literature[31], and bedaquiline levels were substantially lower, but in line with the low CNS levels noted in published studies[32,33]. Importantly, we also demonstrate discordant antibiotic concentrations in the brain parenchyma and CSF compartments, which are likely influenced by drug properties. For example, linezolid, an oxazolidinone with low protein-binding that was found to be beneficial in retrospective studies for the treatment of TB meningitis[34], had substantially higher CSF concentration, but lower levels were achieved in the brain tissues. Conversely, both bedaquiline (and M2 metabolite) as well as pretomanid, which are lipophilic and with higher protein-binding, showed higher concentration in the brain tissue compared with the CSF. This is consistent with differential partitioning of drugs into the protein and lipid rich brain parenchyma versus CSF which is hydrophilic and low in protein. We have noted similar discordance between the CSF and the brain tissue compartments for rifampin and delamanid (also a nitroimidazole, with physiolochemical properties similar to pretomanid) levels[17,20,35], and markers of inflammation[20]. The current studies extend this concept demonstrating that drug properties also influence discordant penetration into the CNS.

The ability to perform cross-species and translational clinical studies with $^{18}$F-pretomanid PET is a major advantage of this

technology. Therefore, we developed a cGMP-compliant radiosynthesis scheme for $^{18}$F-pretomanid to facilitate clinical translation and perform first-in-human imaging studies. First-in-human, dynamic $^{18}$F-pretomanid PET in healthy controls confirmed the excellent but heterogenous brain penetration visualized in our mammalian models of TB meningitis, validating $^{18}$F-pretomanid PET as a powerful translational tool in drug development and treatment optimization. Additionally, the $^{18}$F-pretomanid PET signal in the CSF (brain ventricles), was significantly lower than in other regions of the brain parenchyma. These results confirm compartmentalized and discordant antibiotic penetration into the CNS and highlight the importance of using animal models and imaging technology in humans that can measure antibiotic exposures in the brain parenchyma as well as the CSF.

Our studies have some limitations which we aimed to overcome. Since even minor modifications in the chemical structure of antimicrobials can lead to drastic changes in the physicochemical properties and biological activity of small molecule drugs, we developed a radiosynthetic approach using radiofluorination chemistry that retains the chemical identity of pretomanid. Additionally, the PET studies administered microdoses (ng-µg) of $^{18}$F-pretomanid per subject for the PET studies and current evidence suggests that microdosing is a reliable predictor of the drug biodistribution at therapeutic doses[36,37]. Another limitation of the study is that mass spectrometry drug levels were measured at T$_{max}$ (time to reach maximum concentration) and represent a single time point, and not time-concentration curves

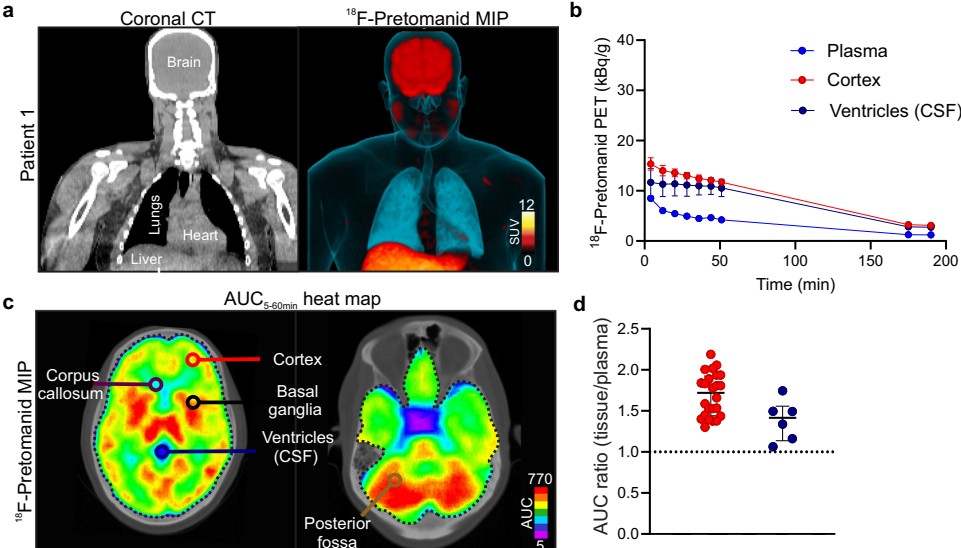

**Fig. 5 | First-in-human $^{18}$F-pretomanid PET. a** Coronal CT and $^{18}$F-pretomanid PET maximum-intensity projection (MIP) images. **b** Representative time-activity curves (TACs) from subject 1 in brain parenchyma and CSF (ventricles) using automated segmentation. A total of 45 volume of interests (VOIs) were analyzed from six subjects. Remaining TACs are shown in Fig. S8. **c** Transverse $^{18}$F-pretomanid PET AUC heat maps demonstrating spatially compartmentalized distribution. **d** AUC$_{4-55min}$ (tissue/plasma) ratios from the cortex and CSF (ventricles) ($P = 0.018$). PET studies are based on microdoses (ng-μg) administered intravenously from six human subjects. SUV, standard uptake values. Data are represented as median ± IQR. Statistical comparisons were performed using 2-way ANOVA followed by Bonferroni's multiple-comparison test. Source data are provided as a Source Data file.

(AUC) as measured by PET. This difference in measurement could explain why the brain/plasma ratios were found to be higher by mass spectrometry compared to PET-derived AUC ratios. Moreover, since $^{18}$F-pretomanid is administered intravenously, the time of injection corresponds to plasma T$_{max}$, and brain uptake reaches C$_{max}$ (maximum concentration) within the first 15 min. Therefore, the first 60 min represent the maximum concentration of $^{18}$F-pretomanid to reach the brain. While rabbit studies utilized both males and females, only female mice were used, and future studies with both males and females would be required to investigate how gender may affect pretomanid PK. Finally, while the reduced penetration of linezolid and bedaquiline into the brain parenchyma are a likely cause of the reduced efficacy of the BPaL regimen compared to standard TB therapy (HR$_{10}$Z), other factors may be considered, and further research is needed. For example, it is possible that bacteria in this model are more susceptible to HR$_{10}$Z due to the effect of isoniazid against fast-replicating bacteria. Furthermore, a dose-escalating study would be useful to assess if increased doses of linezolid and/or bedaquiline might improve outcomes, as we previously observed with high-dose rifampin. In this case however, safety concerns regarding higher doses of linezolid and bedaquiline could be limiting, and analogs could be tested instead.

Here we show the value added by imaging technologies for the characterization of novel drugs and treatment optimization. The use of radiolabeled antimicrobials, combined with conventional microbiology techniques and novel animal models, offers unmatched potential to noninvasively assess PK profiles in infected tissues, characterized by complex, heterogeneous lesions existing simultaneously in the same host[27]. Importantly, our studies demonstrate that pretomanid has excellent penetration into the CNS, confirmed in first-in-human $^{18}$F-pretomanid studies, suggesting that novel pretomanid-based regimens in combination with other antibiotics active against MDR strains and with excellent brain penetration should be considered for the treatment of MDR-TB meningitis. Finally, our studies reaffirm the compartmentalized and discordant antibiotic penetration into the CNS, which are related to the physiochemical properties of the antibiotic. Overall, this is an important finding as CSF studies are commonly utilized in many

clinical trials, but CSF may not be an adequate surrogate of disease or drug levels in the brain tissues.

## Methods

### Study design

The overall goal of this study was to develop radiolabeled-pretomanid for dynamic, concentration-time profiles using PET to characterize pretomanid brain PK (in animal and human studies) and assess the efficacy of a pretomanid-containing regimen for TB meningitis. We performed cross-species analyses for preclinical validation of $^{18}$F-pretomanid as a molecular imaging tool to noninvasively assess whole-body drug penetration and then optimized the radiosynthesis of $^{18}$F-pretomanid under cGMP to facilitate the execution of first-in-human studies. Dynamic $^{18}$F-pretomanid PET/CT was used to obtain TACs and AUCs by quantifying the PET signal in VOIs in animal models and human studies. In animal models, post-mortem autoradiography/histology and mass spectrometry were also performed to characterize pretomanid's heterogenous brain penetration, both spatially and temporally. Given the unknown potential of pretomanid-containing regimens for TB meningitis, BPaL, the only FDA approved pretomanid-containing regimen, was tested in the mouse model of TB meningitis to evaluate intraparenchymal drug levels and bactericidal activity longitudinally. All protocols were approved by the Johns Hopkins University Biosafety, Radiation Safety, Animal Care and Use (RB19M417, MO19M382) and Institutional Review Board Committees (IRB00303845). $^{18}$F-Pretomanid was used according to the FDA Radioactive Drug Research Committee guidelines for investigational drugs.

### Radiosynthesis of $^{18}$F-pretomanid

Radiosynthesis of $^{18}$F-pretomanid was first developed as a manual synthesis using silver-catalyzed nucleophilic displacement for preclinical animal studies. For clinical translation, the radiosynthesis was adapted to an automated cGMP synthesis of $^{18}$F-pretomanid under microwave irradiation in dimethylformamide in the absence of silver salts. Details of the synthesis of $^{18}$F-pretomanid, including the radiolabeling precursor and intermediates and in vitro characterization of $^{18}$F-pretomanid, is available in supplementary materials.

## Animal studies

**Mouse model of pulmonary TB.** Four to six-week-old female C3HeB/FeJ mice (Jackson Laboratory) were aerosol-infected with frozen stocks of *M. tuberculosis* (H37Rv), using the Middlebrook Inhalation Exposure System (Glas-Col)[38].

**Mouse model of TB meningitis.** Female C3HeB/FeJ mice (7–8 weeks old, Jackson laboratories) were infected intraventricularly (titrated frozen stocks with ~6.5 $\log_{10}$ CFU of *M. tuberculosis* H37Rv) via a burr hole using a Hamilton syringe (Hamilton, 88,000) and stereotaxic instrument (David KOPF instrument, model 900, coordinates 0.6 mm dorsal to bregma, 1.2 mm lateral to middle line and 2 mm ventral)[20]. The infection was allowed to incubate for 14 days.

**Rabbit model of TB meningitis.** Male and female New Zealand White rabbits (5–7 day old, Robinson Services Inc.) were infected intraventricularly (titrated frozen stock with ~6.5 $\log_{10}$ of *M. tuberculosis* H37Rv) via the bregma using a 30-gauge insulin syringe. Prior to infection, rabbits were sedated with dexmedetomidine hydrochloride (0.2 µg/g; Zoetis, Florham Park, NJ) and topical anesthesia (lidocaine 4%; Ferndale IP Inc., Ferndale, MI) was applied to the bregma[17,19,20]. The infection was allowed to incubate for at least 14 days before imaging was performed.

**Ethical approval.** All the animal experiments in this study were approved by the Johns Hopkins University Animal Care and Use Committee.

**Ex vivo biodistribution.** Anesthetized mice ($n = 4$ per time point) were injected with [18]F-pretomanid ($2.6 \pm 1.9$ MBq) via the tail vein and sacrificed 1, 2, and 4 h post-injection. Organs and tissues of interest were harvested, weighed, and the radioactivity gamma-counted. Data are represented as percent injected dose per weight of tissue (%ID/g).

**Autoradiography.** Following intravenous injection of [18]F-pretomanid, animals were sacrificed, perfused with saline, and brains collected. Tissues were embedded in OCT and 20 µm sections placed on slides. The radioactive slides were placed inside an exposure cassette (GE, code no. 29175523) and exposed for 2–5 half-lives before being developed by phosphor storage imaging (GE Typhoon). Following radioactive decay, the tissues used for autoradiography were fixed in formalin and underwent hematoxylin and eosin staining.

**Antimicrobial treatments and efficacy.** Drug stocks were prepared and administered five days a week as previously described[30,39]. Rifampin (10 mg/kg/day), isoniazid (10 mg/kg/day), pyrazinamide (150 mg/kg/day), pretomanid (50 mg/kg/day divided twice daily), bedaquiline (25 mg/kg/day), and linezolid (100 mg/kg/day divided twice daily) were prepared for oral administration via gavage and dexamethasone was prepared for intraperitoneal injection and stored at 4 °C. Mouse dosing was utilized to match the standard human equipotent dosing: rifampin (10 mg/kg/day), isoniazid (10 mg/kg/day), pyrazinamide (25 mg/kg/day), pretomanid (200 mg/day)[40], bedaquiline (standard oral dosing)[41], and linezolid (1200 mg/day)[42]. Treatment efficacy was determined by whole brain bacterial burden (CFU) at 2 and 6 weeks after initiation of treatment using 7H11 plates supplemented with activated charcoal.

**Imaging.** *M. tuberculosis*-infected mice and rabbits were imaged inside transparent and sealed biocontainment containers, compliant with biosafety level (BSL)-3 containment and capable of delivering $O_2$-anesthetic mixture to sustain live animals during imaging as previously described[43]. PET acquisition and subsequent CT were performed using the nanoScan PET/CT (Mediso, Arlington, VA) and images automatically co-registered. [18]F-Pretomanid PET/CT: Mice were injected with [18]F-pretomanid (pulmonary TB, $3.77 \pm 1.78$ MBq; TB meningitis, $3.88 \pm 1.44$ MBq) via the tail vein. Anesthetized rabbits were injected with [18]F-pretomanid ($4.65 \pm 0.75$ MBq) through the ear vein. The injection time coincided with the start of the dynamic PET acquisition. [18]F-FDG PET/CT: Animals were imaged as previously described[20].

**Mass spectrometry.** Terminal samples (blood, brain and CSF) were collected at the appropriate plasma $T_{max}$ for each antimicrobial[44–46]. Blood was collected in EDTA tubes (BD Microtainer, Fisher Scientific) for plasma separation. Bedaquiline, bedaquiline M2 metabolite, pretomanid and linezolid in plasma, CSF and brain tissues from *M. tuberculosis*-infected mice were quantified using validated ultra-high-performance liquid chromatography (UPLC) and tandem mass spectrometry (LC–MS/MS) at the Infectious Diseases Pharmacokinetics Laboratory of the University of Florida. Calibration standard curves ranges were bedaquiline (and M2 metabolite) 2.00 to 0.01 µg/mL, pretomanid 30.00 to 0.01 µg/mL, and linezolid 30.00 to 0.03 µg/mL.

**Cytokine analysis.** Samples for cytokine analysis were collected from supernatants after centrifuged brain homogenization and stored at −80 °C until analysis. IL-1α, vascular endothelial growth factor A (VEGFA) and matrix metalloproteinase-8 (MMP-8) were analyzed on Luminex Multiplex assays by the Oncology Human Immunology Core of Johns Hopkins University.

## Human studies

[18]F-Pretomanid was synthesized as a sterile solution with high specific activity ($40.27 \pm 12.82$ GBq/µmol) and high radiochemical purity (>95%) using cGMP by the Johns Hopkins PET Radiotracer Center. Written informed consent was obtained from all healthy volunteers and deidentified images were analyzed. All subjects had a physical exam by a trained physician and screening laboratory tests before imaging to confirm eligibility. Each subject received an intravenous bolus of $359.52 \pm 2.79$ MBq of [18]F-pretomanid followed by dynamic PET utilizing a multi-bed protocol immediately after tracer injection (0–60 min) and 180 min after tracer injection (180–210 min) using Siemens Biograph mCT 128-slice scanner. All subjects were assessed for adverse events immediately after the completion of the imaging studies and via a follow up telephone interview at 20–25 days after the imaging studies. A trained radiologist also evaluated the CT images for all subjects to assess for any anatomic abnormalities.

## Image analysis

PET/CT images were analyzed using VivoQuant 2020 (Invicro) or PMOD (3.402) for animal and human studies, respectively. In humans, the brain segmentation tool (Hammers N30R83) was used to draw VOIs and OsiriX MD 11.0 DICOM Viewer (Pixmeo SARL) was used to create 3D MIP images. Data for blood were obtained by placing a VOI in the left heart ventricle and then correcting to plasma using individual subject hematocrit values or standard hematocrit[47] and red blood cell partition coefficient of pretomanid[48] (Table S5). PET data were adjusted for mass using the density of each organ obtained from the CT (Hounsfield units). Data are expressed as %ID/g or standard uptake values (SUV) in animal or human studies, respectively. Heatmap overlays were created using RStudio.

## Statistical analysis

Data were analyzed using Prism 8 version 8.1.0 (GraphPad). AUCs were calculated using the linear trapezoidal rule. Comparisons were made using a two-tailed, Mann–Whitney-Wilcoxon test or ANOVA followed by Bonferroni's multiple-comparison test. $P$ values $\leq 0.05$ were considered statistically significant.

## Reporting summary

Further information on research design is available in the Nature Portfolio Reporting Summary linked to this article.

## Data availability

All data are available in the main text or the supplementary materials. Source data are provided with this paper.

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

## Acknowledgements

The authors would like to thank Eric Nuermberger (Johns Hopkins Hospitals) for assistance with antibiotic dosing, Mariah Klunk for assistance with PET/CT imaging, and Axia Chemicals Ltd. for developing the precursor for [18]F-pretomanid. We also want to thank all the study subjects as well as Amanda Henderson, Ergi Spiro and Jeff Leal (Johns Hopkins Hospitals) for help with recruiting the human subjects and curating the human imaging data respectively. Funding: This work was funded by the US National Institutes of Health R01-AI145435-A1 (S.K.J.), R01-AI153349 (S.K.J.), R01-HL131829 (S.K.J.), R21- AI149760 (S.K.J.), and K08-AI139371 (E.W.T.).

## Author contributions

F.M., C.A.R.-B. and S.K.J. conceptualized and designed the studies. F.M. and P.D.J. performed the manual radiosyntheses. D.P.H. and R.F.D. developed the cGMP synthesis. F.M., C.A.R.-B., P.D.J., M.B. and K.F. performed the mouse studies. X.C. performed the cytokine analysis. F.M., C.E., J.K. and E.W.T. performed the rabbit studies. E.W.T. supervised the rabbit studies. A.A.O. and S.K.J. wrote the protocol for the human studies. A.A.O., C.A.R.-B., E.W.T. and M.K.B. recruited and consented the human subjects and analyzed the human imaging data. C.A.P. performed mass spectrometry analysis. S.K.J. provided funding and supervised the project. F.M., C.A.R.-B., E.W.T. and S.K.J. wrote the manuscript with substantial input from all co-authors.

## Competing interests

The authors declare no competing interests.

## Additional information

[1]Center for Infection and Inflammation Imaging Research, Johns Hopkins University School of Medicine, Baltimore, MD 21287, USA. [2]Center for Tuberculosis Research, Johns Hopkins University School of Medicine, Baltimore, MD 21287, USA. [3]Department of Pediatrics, Johns Hopkins University School of Medicine, Baltimore, MD 21287, USA. [4]Department of Anesthesiology and Critical Care Medicine, Johns Hopkins University School of Medicine, Baltimore, MD 21287, USA. [5]Russell H. Morgan Department of Radiology and Radiological Sciences, Johns Hopkins University School of Medicine, Baltimore, MD 21287, USA. [6]Infectious Disease Pharmacokinetics Laboratory, Pharmacotherapy and Translational Research, University of Florida College of Pharmacy, Gainesville, FL 32610, USA. [7]These authors contributed equally: Filipa Mota, Camilo A. Ruiz-Bedoya, Elizabeth W. Tucker, Daniel P. Holt. ✉e-mail: sjain5@jhmi.edu

