## [Peer review file · Nature Communications]

REVIEWER COMMENTS

Reviewer #1 (Remarks to the Author):

This manuscript describes studies in mice, rabbits and humans which define the penetration of pretomanid into the brain and the potential activity of the BPAL regimen for the treatment of tuberculous meningitis. The studies address clinically important questions for which there are few data either from animals or humans.

In general, the manuscript is well written, the study designs are appropriate and clearly described, and the conclusions balanced. I have no major criticisms and congratulate the authors for the high quality of their work.

My comments are therefore relatively minor, but I have two suggestions:

1. In figure 4 E the AUC ratios for radiolabelled Bedaquiline and Linezolid are presented, yet it is apparent in an asterixed footnote that these data do not derive from the current experiments. In my view, it is confusing, and potentially misleading, for previously published data to be presented within the results section, especially without any details within the paper on the methods used to acquire those data. These previous data are clearly important and relevant but I suggest removing them to the discussion, where the results are placed in broader context.
2. The lack of intracerebral activity of the BPAL regimen compared to RHZ is a challenging finding with many possible explanations. Whilst I have no concern about the data themselves, the limitations of the experiment, and therefore the strength of the conclusions, could be explored in greater depth in the discussion. These include the relatively short duration of the experiments, the likely major contributions of isoniazid to early killing in the standard regimen, the desirability of adding a fluoroquinolone to BPAL to enhance early bactericidal activity in CNS TB, and whether H37Rv is the best choice of bacterial strain for these experiments. I think the authors have a responsibility to provide a very balanced view here, that mitigates the risk the regimen is discarded prematurely for the treatment of tuberculous meningitis before essential clinical data can be acquired.

Reviewer #2 (Remarks to the Author):

The authors' Mota et al have developed an ¹⁸F synthesis and labeling scheme for Pretomanid (¹⁸Fpa) that maintains very similar physiochemical properties to the original agent Pa. They use this ¹⁸F agent to study the distribution and kinetics of the compound in two Tuberculosis (TB) models in the mouse (pulmonary and meningitis) and the rabbit meningitis model. The results show that the agent reaches the lung and crosses the blood-brain barrier sufficiently that it significantly exceeds plasma levels in

brain tissue. These results are shown by both PET/CT imaging methods and by HPLC of tissues after sacrifice. They go on to make a cGMP synthesis route for the ^{18}F Pa and show that the probe crosses the blood-brain barrier in healthy humans also. The concentrations observed in the brain were shown to be compartmentalized or region-specific within the brain. During their investigation of the recently WHO-advocated regimen BPaL for MDR-TB, the group showed that the BPaL regimen that is effective against pulmonary TB in mice is much less so in meningitis and significantly less active than the standard HRZ regimen. This is a significant finding that deserves both attention from those treating meningitis and more exploration by the scientific community. Finally, the authors assert that their approach (producing a PET probe of anti-infectives to study the compound's in vivo behavior) provides significant insight into drug behavior.

I mostly have only minor comments that will improve the readers understanding of the paper but I do have a couple of thoughts of importance.

1) The mouse meningitis model used here only allowed the disease to establish over two weeks. This may not have been enough time for the host-bacteria interaction to generate lesions that restrict the growth rate of the bacteria to render them into a persistent or slowly growing state that is thought to be less susceptible to INH. It is certainly likely that the difference in activity is due to the lower levels of BDQ and LIN in that environment rendering this single-drug treatment. What is thought to be the state of MTB bacteria in human meningitis? Is this an adequate model? What is the contribution of the metabolic state of bacteria in this model? Are they still more susceptible to the INH containing standard treatment than a longer model would be.

2) What are the drug concentrations once a steady state is reached? While this would be hard to do in an infected model, is steady state for BDQ reached after 6 weeks?

3) I'd suggest that the authors be slightly repetitive and be sure that the figure legends indicate if the measures reported are single dose or steady state measures. Of course the PET/CT results are going to be sub-therapeutic and single dose results – but putting the dosing data and timing into the figures would be helpful as is requested below in the more minor comments.

Minor comments

Abstract:

“no well-accepted” treatments – consider rewording this sentence. It is the MDR resistance that results in high mortality. Lack of highly efficacious treatments, perhaps.

the low activity of BPAL was said to be “likely due to restricted penetration of bedaquiline and linezolid into the brain parenchyma”. That suggests to me that the HRZ regimen components enter the CSF and brain tissue. I think JHU has published data related to this earlier – so perhaps this could be reworded to contrast this with drugs from the standard regimen. Did all of those reach therapeutic levels.

Also this manuscript seems to indicate that Pa has low penetration into lesions (at least by radiography). So, do the authors think this also contributes? It seems like further consideration of the message of this sentence is warranted. Was the concentration within lesions too low to be therapeutic?

The limited time of penetration for ^{18}F Pa in the mouse and rabbit PET/CT's may underrepresent the final penetration of Pa into the brain and lesions over time. The long $T_{1/2}$ of Pa in plasma would usually indicate that assessment at only 1 hour might not reflect the whole picture. Adding this more clearly to the limitations is suggested if you don't have data to the contrary.

Fig4 graph E (AUC needs a time range, 0-4?) to be evaluated. Graphs F, G, H need time point for the presented data and if the data are from animals at single dose or steady state.

Fig5 Panel A, if the max and min could be replaced with SUV 0 to X, that would be preferred unless the PET is representing the AUC of the exposure. If so, this should be added to the figure legend. Although the AUC of the ^{18}F Pa was different within the hour across several of the brain regions, it is unclear if this difference is thought to be clinically relevant. Thoughts on this should be added to the discussion.

Discussion: top of page 15. “Pa remains high even after Dex treatment”. I might have missed this, but it's not clear which figure presents this later in treatment penetration data. Is it only S6 where the data are from 2 weeks? Were these effective doses so that any change in the fidelity or nature of the blood-brain barrier would have been observed?

Delamanid shares the ability to cross the blood-brain barrier as I remember. If it is right, – I think I remember seeing it from JHU also, then mentioning it in the discussion might be worthwhile – is it the nature of nitroimidazoles?

Figure S4 A. There is a concentration of ^{18}F Pa activity above the liver in the animal shown in A at 180 minutes. Is that structure the heart? And if so, might it be labeled? B. How was the blood activity represented in this figure? Is it adjusted for the total volume of blood? Is bladder wall data available or could it be noted in the legend text. A, C-E are 0 to 3 hours; but B is 1, 2, and 4 hours? Is this correct?

Fig S4F I'd suggest adding the conclusion to the legend since it is not at all obvious. May also be worthwhile marking the CT lesion in the histology or autoradiography panel.

Figure S5. A simple conclusion would be helpful here also.

Table S2. Please add N= for the various groups in the legend or in the methods.

Table S3. Since the data are median concentrations rather than AUCs; it would be good to provide the time point for the reader, even if they are different for different drugs. Also since the half-life of these 3 agents are so different, is the data available to provide AUC or C_{avg} ? For BDQ and Pa, C_{avg} are useful measures, while AUC would be better for Linezolid.

Reviewer #3 (Remarks to the Author):

The manuscript by Mota et al describes work where the investigators developed a ^{18}F -labeled version of the anti-mycobacterial drug pretomanid so that drug penetrance during treatment could be measured in vivo by PET-CT. The work is specifically targeted at TB meningitis, which is a significant problem with poor outcomes and limited treatment modalities. The authors demonstrated that they were able to generate the labeled product, and do it in a GMP process that enabled pretomanid uptake to be tested in healthy volunteers, as well as to be tested in murine and rabbit models of TB meningitis. In addition to the ^{18}F -labeled pretomanid, the authors tested pretomanid in combination therapy with radiolabeled bedaquiline and linezolid to compare uptake of these three drugs.

The investigators found that there was good penetrance of ^{18}F -pretomanid past the blood-brain barrier but limited penetrance into granulomas while ^{18}F -FDG took was taken up well by these lesions. This is not surprising because of the nature of these lesions and this is useful information on the behavior of this drug. The BPaL treatment regimen had limited success in reducing bacteria loads when compared to the standard regimen. The investigators attributed this outcome to poor penetrance of the non-pretomanid drugs in this cocktail, which is supported by the data they present. These are useful data and could inform development and optimization of treatment regimens for TB meningitis.

The manuscript is well written and the materials and methods in the main manuscript and in the supplemental data section appear to be detailed enough to evaluate the study and repeat the work. Note: this reviewer does not have the background to assess the synthesis process but the results appear to yield appropriate data and are presented in a way that is consistent with other studies in this arena. The Mtb infection work and outcomes are described in enough detail to be evaluated and appear to be appropriately performed. The authors included a paragraph that describes the limitation of the study in the Discussion that addresses some weaknesses of the study. The strength of the data and the results are appropriately framed.

The findings are somewhat negative because of the lack of the treatment regimen's lack of efficacy but the approach and data are useful and have translational value for treatment of this serious problem. No major or substantial concerns were noted in the manuscript.

RESPONSE TO REVIEWERS: We thank the reviewers for evaluating our original manuscript and making suggestions that further strengthen it. Each comment has been addressed, and a point-by-point response to all comments is listed.

REVIEWER 1: This manuscript describes studies in mice, rabbits and humans which define the penetration of pretomanid into the brain and the potential activity of the BPaL regimen for the treatment of tuberculous meningitis. The studies address clinically important questions for which there are few data either from animals or humans. In general, the manuscript is well written, the study designs are appropriate and clearly described, and the conclusions balanced. I have no major criticisms and congratulate the authors for the high quality of their work.

Author response: We thank the reviewer for the positive comments. Specific comments are addressed below:

1. **Reviewer comment:** In figure 4 E the AUC ratios for radiolabelled Bedaquiline and Linezolid are presented, yet it is apparent in an asterixed footnote that these data do not derive from the current experiments. In my view, it is confusing, and potentially misleading, for previously published data to be presented within the results section, especially without any details within the paper on the methods used to acquire those data. These previous data are clearly important and relevant but I suggest removing them to the discussion, where the results are placed in broader context.

Author response: We appreciate this comment and in response, we have moved the data derived from these previously published studies (Fig. 4E) to supplementary data (Fig. S6).

2. **Reviewer comment:** The lack of intracerebral activity of the BPal regimen compared to RHZ is a challenging finding with many possible explanations. Whilst I have no concern about the data themselves, the limitations of the experiment, and therefore the strength of the conclusions, could be explored in greater depth in the discussion. These include the relatively short duration of the experiments, the likely major contributions of isoniazid to early killing in the standard regimen, the desirability of adding a fluoroquinolone to BPaL to enhance early bactericidal activity in CNS TB, and whether H37Rv is the best choice of bacterial strain for these experiments. I think the authors have a responsibility to provide a very balanced view here, that mitigates the risk the regimen is discarded prematurely for the treatment of tuberculous meningitis before essential clinical data can be acquired.

Author response: This is an interesting point and we have added this as a limitation in the discussion. Specifically, we have added, “Finally, while the reduced penetration of linezolid and bedaquiline into the brain parenchyma are a likely cause of the reduced efficacy of the BPaL regimen compared to standard TB therapy (HR₁₀Z), other factors may be considered, and further research is needed. For example, it is possible that bacteria in this model are more susceptible to HR₁₀Z due to the effect of isoniazid against fast-replicating bacteria. Furthermore, a dose-escalating study would be useful to assess if increased doses of linezolid and/or bedaquiline might improve outcomes, as we previously observed with high-dose rifampin. In this case however, safety concerns regarding higher doses of linezolid and bedaquiline could be limiting, and analogs could be tested instead.” Although treatment for 6-weeks may be relatively short, differences in bedaquiline containing treatment regimens for pulmonary TB have been noted within 6-weeks of treatment initiation.

REVIEWER 2: We thank the reviewer for their comments and suggestions. Specific comments are addressed below:

1. **Reviewer comment:** The mouse meningitis model used here only allowed the disease to establish over two weeks. This may not have been enough time for the host-bacteria interaction to generate lesions that restrict the growth rate of the bacteria to render them into a persistent or slowly growing state that is thought to be less susceptible to INH. It is certainly likely that the difference in activity is due to the lower levels of BDQ and LIN in that environment rendering this single-drug treatment. What is thought to be the state of MTB bacteria in human meningitis? Is this an adequate model?

What is the contribution of the metabolic state of bacteria in this model? Are they still more susceptible to the INH containing standard treatment than a longer model would be.

Author response: This is an interesting point and we acknowledge that this issue is not easily answerable. However, and in response to this comment, we have added this to the discussion under limitations (see Reviewer 1, point 2 above).

- 2. Reviewer comment:** What are the drug concentrations once a steady state is reached? While this would be hard to do in an infected model, is steady state for BDQ reached after 6 weeks?

Author response: The plasma half-life of bedaquiline and its M2 metabolite in C3HeB/FeJ mice (used in the current study) is 116.4 and 55.9 hours respectively (Irwin *et al.* ACS Infect Dis. 2016). Similarly, the half-life of bedaquiline and its M2 metabolite in pulmonary TB lesions in C3HeB/FeJ mice is 104.4 and 98.6 hours respectively (Irwin *et al.* ACS Infect Dis. 2016). Although these data are not from the CNS compartment, even if we use the longest half-life estimate (116.4 hours), steady state should be achieved in 24.2 days (5 half-lives), which is well within the 6-weeks mark.

- 3. Reviewer comment:** I'd suggest that the authors be slightly repetitive and be sure that the figure legends indicate if the measures reported are single dose or steady state measures. Of course the PET/CT results are going to be sub-therapeutic and single dose results – but putting the dosing data and timing into the figures would be helpful as is requested below in the more minor comments.

Author response: We appreciate this comment and have added this information to the figure legends (Figs. 3-5, S4-S6, S8).

MINOR COMMENTS

- 1. Reviewer comment:** Abstract: “no well-accepted” treatments – consider rewording this sentence. It is the MDR resistance that results in high mortality. Lack of highly efficacious treatments, perhaps.

Author response: We have changed this to, “. . . there is a lack of efficacious treatments for TB meningitis due to MDR strains.”

- 2. Reviewer comment:** Abstract: the low activity of BPaL was said to be “likely due to restricted penetration of bedaquiline and linezolid into the brain parenchyma”. That suggests to me that the HRZ regimen components enter the CSF and brain tissue. I think JHU has published data related to this earlier – so perhaps this could be reworded to contrast this with drugs from the standard regimen. Did all of those reach therapeutic levels.

Author response: We (and others) have previously demonstrated that rifampin has restricted penetration into the CNS, which can be overcome by using higher doses. Limited penetration of bedaquiline into the CSF is already known but brain penetration of all three drugs (BPaL) is not well-established. We have now clarified this in the abstract.

- 3. Reviewer comment:** Also this manuscript seems to indicate that Pa has low penetration into lesions (at least by radiography). So, do the authors think this also contributes? It seems like further consideration of the message of this sentence is warranted. Was the concentration within lesions too low to be therapeutic?

Author response: The AUC ratio (brain/plasma) are indeed lower in the brain lesions compared to the unaffected brain but still higher than 1 and so likely to be therapeutic. We have clarified this in the discussion.

4. **Reviewer comment:** The limited time of penetration for ¹⁸F-Pa in the mouse and rabbit PET/CT's may underrepresent the final penetration of Pa into the brain and lesions over time. The long T_{1/2} of Pa in plasma would usually indicate that assessment at only 1 hour might not reflect the whole picture. Adding this more clearly to the limitations is suggested if you don't have data to the contrary.

Author response: This is an interesting point and we have added this to the discussion under the limitation section. ¹⁸F-Pretomanid PET imaging in humans was performed up to 210 min (3.5 hours) and showed that concentration (kBq/g) into the brain remains higher than in plasma at the final time point. While ¹⁸F-pretomanid PET imaging in mice and rabbits were performed for only 60 minutes post-injection, our data show that brain concentration in mice and rabbits also remains higher than in plasma. Since ¹⁸F-pretomanid is administered intravenously, the time of injection corresponds to plasma T_{max}, and brain uptake reaches C_{max} within the first 15 min. Thus, the first 60 minutes represent the maximum concentration of ¹⁸F-pretomanid to reach the brain.

5. **Reviewer comment:** Fig4 graph E (AUC needs a time range, 0-4?) to be evaluated. Graphs F, G, H need time point for the presented data and if the data are from animals at single dose or steady state.

Author response: We thank the reviewer for pointing this out and have added this information to the figure legend.

6. **Reviewer comment:** Fig5 Panel A, if the max and min could be replaced with SUV 0 to X, that would be preferred unless the PET is representing the AUC of the exposure. If so, this should be added to the figure legend. Although the AUC of the ¹⁸F-Pa was different within the hour across several of the brain regions, it is unclear if this difference is thought to be clinically relevant. Thoughts on this should be added to the discussion.

Author response: We thank the reviewer for pointing this out and have added the SUV values for panel A. Additionally, we agree about the limited clinical relevance of the different brain areas and have simplified the graphs. Similar changes were made to Fig. S8.

7. **Reviewer comment:** Discussion: top of page 15. "Pa remains high even after Dex treatment". I might have missed this, but its not clear which figure presents this later in treatment penetration data. Is it only S6 where the data are from 2 weeks? Were these effective doses so that any change in the fidelity or nature of the blood-brain barrier would have been observed?

Author response: Both regimens (HR₁₀Z or BPaL) included adjunctive dexamethasone treatment at doses that indeed have measurable therapeutic effects (Ruiz-Bedoya *et al.* J Clin Invest. 2022). The treatments were administered for two weeks (now added to the figure legend) when changes to the blood-brain barrier permeability are evident as demonstrated by us previously (Ruiz-Bedoya *et al.* J Clin Invest. 2022).

8. **Reviewer comment:** Delamanid shares the ability to cross the blood-brain barrier as I remember. If it is right, – I think I remember seeing it from JHU also, then mentioning it in the discussion might be worthwhile – is it the nature of nitroimidazoles?

Author response: As suggested, we have added information on delamanid to the discussion. However, the nitroimidazole moiety is a relatively small component of the molecules themselves, and there is not enough data in the literature to allow us to conclude that all nitroimidazoles will equate to good brain penetration.

9. **Reviewer comment:** Figure S4 A. There is a concentration of ^{18}F Pa activity above the liver in the animal shown in A at 180 minutes. Is that structure the heart? And if so, might it be labeled? B. How was the blood activity represented in this figure? Is it adjusted for the total volume of blood? Is bladder wall data available or could it be noted in the legend text. A, C-E are 0 to 3 hours; but B is 1, 2, and 4 hours? Is this correct?

Author response: In panel A, the activity shown at 180 min at the top of the liver corresponds to the gallbladder, which has now been labelled. In panel B, blood was collected directly from the heart and was corrected to plasma based on RBC partition coefficient and hematocrit. In all panels of Figure S4 the measurements represent a concentration of percent injected dose per gram of tissue (%ID/g) and this is generally assumed to be representative of the whole tissue thus not requiring adjustment to total blood volume. In the PET-derived data the bladder concentration represents the bladder wall and contents as it is not possible to distinguish between them. In the *ex vivo* data (panel B), bladder wall data was not collected. It is correct that panels A, C-E (in vivo PET) correspond to 0 to 3 hours, but panel B (*ex vivo*) corresponds to 1, 2, and 4 hours post-injection.

10. **Reviewer comment:** Fig S4F I'd suggest adding the conclusion to the legend since it is not at all obvious. May also be worthwhile marking the CT lesion in the histology or autoradiography panel.

Author response: We have added a conclusion to panel F highlighting that "autoradiography shows heterogeneous distribution of ^{18}F -pretomanid".

11. **Reviewer comment:** Figure S5. A simple conclusion would be helpful here also.

Author response: We thank the reviewer for this suggestion and have added the following to the legend, " ^{18}F -Pretomanid upper body biodistribution in the rabbits is similar to what is noted in the mouse studies."

12. **Reviewer comment:** Table S2. Please add N= for the various groups in the legend or in the methods.

Author response: There are 5-10 per animals per group in the treatment arms and 4 animals in the untreated group. We have added this to the legend for Table S2.

13. **Reviewer comment:** Table S3. Since the data are median concentrations rather than AUCs; it would be good to provide the time point for the reader, even if they are different for different drugs. Also since the half-life of these 3 agents are so different, is the data available to provide AUC or Cavg? For BDQ and Pa, Cavg are useful measures, while AUC would be better for Linezolid.

Author response: The time-point at which samples were collected has been added to Table S3. Since these concentrations were acquired at a single time point it is not possible to provide mass spectrometry-derived AUC or Cavg values for this experiment.

REVIEWER 3

Reviewer comment: The findings are somewhat negative because of the lack of the treatment regimen's lack of efficacy but the approach and data are useful and have translational value for treatment of this serious problem. No major or substantial concerns were noted in the manuscript.

Author response: We thank the reviewer for the positive comments.